# Measurement Invariance of the WHO-5 Well-Being Index: Evidence from 15 European Countries

**DOI:** 10.3390/ijerph19169798

**Published:** 2022-08-09

**Authors:** Alina Cosma, András Költő, Yekaterina Chzhen, Dorota Kleszczewska, Michal Kalman, Gina Martin

**Affiliations:** 1Department of Sociology, Trinity College Dublin, D01 Dublin, Ireland; 2Health Promotion Research Centre, School of Health Sciences, National University of Ireland Galway, H91 TK33 Galway, Ireland; 3Foundation of the Institute of Mother and Child, 01-211 Warsaw, Poland; 4Department of Recreation and Leisure Studies, Faculty of Physical Culture, Palacký University Olomouc, 779 00 Olomouc, Czech Republic; 5Faculty of Health Disciplines, Athabasca University, Athabasca, AB T9S 3A3, Canada

**Keywords:** mental well-being, mental health, measurement invariance, gender, cross-national, age differences, HBSC

## Abstract

(1) *Background:* The World Health Organization (WHO)-5 Well-Being Index has been used in many epidemiological studies to assess adolescent mental well-being. However, cross-country comparisons of this instrument among adolescents are scarce and, so far, no good-fitting, common invariant measurement model across countries has been reported. The present study aims to evaluate and establish a version of the WHO-5 Well-Being Index that allows for a valid cross-country comparison of adolescent self-reported mental well-being. (2) *Methods:* Using data from the 2018 Health Behaviour in School-aged Children study, we evaluated the measurement model and measurement invariance of the five items of the WHO-5 Well-Being Index. We used nationally representative samples of 11-, 13-, and 15-year-old adolescents (*N* = 74,071) from fifteen countries and regions in Europe. Measurement invariance of the WHO-5 was assessed using a series (country, gender, and age) of multi-group confirmatory factor analyses. In addition, we evaluated the convergent validity of the measure by testing its correlations with psychosomatic complaints, life satisfaction, and self-rated health. (3) *Results:* We found that WHO-5 does not show good psychometric properties or good measurement invariance fit. However, by excluding the first item of the scale (“I have felt cheerful and in good spirits”), the WHO-4, consisting of the other four original items, had good psychometric properties, and demonstrated good suitability for cross-national comparisons (as well as age and gender) in adolescent mental well-being. (4) *Conclusions:* The present study introduces the WHO-4—a revised version of the WHO-5—, that allows for a valid comparison of mental well-being across fifteen countries and regions in Europe. The WHO-4 proved to be a reliable and valid instrument to assess mental well-being in the adolescent population.

## 1. Introduction

We are witnessing a global public health crisis in adolescent mental health [1], manifesting in increasing mental health problems [2,3]. Therefore, adolescent mental health has become one of the main priorities of research, policy, and health and social services. The World Health Organization (WHO) defines mental health as “a state of well-being in which the individual realises his or her own abilities, can cope with the normal stresses of life, can work productively and fruitfully, and is able to make a contribution to his or her own community” [4]. This conceptualisation implies that mental health includes positive functioning (or mental well-being) as well as problems or illness. Thus, for the purpose of this investigation, we use the term *mental well-being* to refer to an overall positive mental state that we want to measure. Assessment of mental well-being among adolescents requires instruments that are age-appropriate and acceptable for that population [5]. Therefore, testing for measurement invariance (MI)—i.e. a statistical property of measurement that indicates that the same latent construct is being measured across specific groups (e.g., gender, age, ethnicity, countries)—becomes essential, especially in cross-cultural studies of health. Demonstrating measurement invariance is paramount for survey measures that are intended to be used across various groups and cultures. The aim of this paper is to test the MI of the WHO-5 Well-Being Index among adolescents across countries, genders, and age groups.

### 1.1. The WHO-5 Well-Being Index

The five-item World Health Organization Well-Being Index (WHO-5) [6,7] is one of the most widely used measures of mental well-being in clinical and population-based studies. Individuals are asked to indicate on a six-point Likert scale ranging from 0 (“at no time”) to 5 (“all of the time”) how they felt in the past two weeks in terms of their mood (e.g., feeling calm and relaxed, or waking up fresh and rested). This instrument was originally developed as a measure of mental well-being for adult populations [7]. Subsequently, it was validated as a screening tool for depression both in adult and adolescent samples [6] and started to be used in populational-based surveys with adolescents, e.g., [8]. 

The WHO-5 has been validated in a number of studies with regard to both clinical and psychometric validity, e.g., [9]. A systematic review of the WHO-5 (including studies on children and adolescents) concluded that “WHO-5 is a short questionnaire consisting of five simple and non-invasive questions, which tap into the mental well-being of the respondents” [6]. Previous studies on adolescent samples report that the WHO-5 has high internal consistency, with a Cronbach-alpha of 0.70 in Gambia [10]; 0.82 in the Netherlands [11]; 0.90 in Malaysia [12]; and 0.91 in Iran [13]. The above-cited studies and many other publications demonstrate that all five items load on one latent factor. The scale has adequate validity both as a screening tool for depression and as an outcome measure in clinical trials, and has been applied successfully across a wide range of study fields [6]. Furthermore, the WHO-5 seems to be suitable for use among children and adolescents in pediatric care (ages 9 to 16), and can be used as a screening instrument for depressive mood [14,15]. 

### 1.2. Convergent/Construct Validity

Previous studies report negative correlations between the WHO-5 and other depression screening instruments [10,11,16,17], whereas a positive correlation has been reported with life satisfaction or other positive well-being outcomes [16,18]. Several studies of adolescent populations used the WHO-5 as an outcome. For example, participation in sports activities and frequency of physical activity were associated with higher mental well-being among European adolescents aged 14 to 16 years [8]. Functional impairment related to poor health was associated directly and indirectly with lower WHO-5 scores in Hungarian adolescents aged 14 to 16 years [19]. Estonian adolescents (grades 5 to 9) experiencing major cyberbullying reported lower scores of WHO-5 than those who had not been cyberbullied [20].

### 1.3. Measurement Invariance

An increasing number of cross-national surveys of adults and adolescents include latent measures of mental well-being with the purpose of undertaking cross-national comparisons. However, latent variable scores can only be meaningfully compared across countries and/or over time if the measurement structures underlying these latent factors are stable (“invariant”) across such groups [21,22,23,24]. As outlined by Romano et al. [25], measurement invariance (MI) means that the given scale measures the same latent construct across subgroups within a sample [26]. The authors argue that establishing MI is a prerequisite for making meaningful group-level comparisons [26]. Measurement of non-invariance, on the other hand, may lead to interpretation biases due to differences between subgroups [27]. In the latter case, observed differences in scores may be artefactual and invalid, and therefore group comparisons are not recommended. Most research that explores measurement invariance tests for configural, metric, and scalar invariance. *Configural invariance* means that the items of a scale have the same factor structure between groups through a multi-group confirmatory factor analytic (CFA) model. *Metric invariance* means that there are equal loadings between groups, which can be interpreted as the factor structure having the same meaning across groups. *Scalar invariance* means that the intercepts of the model are equal, therefore means can be compared meaningfully across groups [27].

To our knowledge, there is only one published study that explored cross-national MI of the WHO-5 [28]. The analysis revealed that the WHO-5 meets the criteria for configural and metric invariance but discarded scalar invariance across countries. However, when fixing item thresholds, non-invariance was indicated, at least for some threshold parameters. It is important to note that the age of the participants included in this study ranged from 15 to 89 years (*M*_age_ = 43.3, SD = 12.7), which means that cross-national investigation of WHO-5 in samples of adolescents are missing from the research literature.

### 1.4. Present Study

The present study aims to evaluate the measurement invariance of the WHO-5 among 11-, 13-, and 15-year-old adolescents across European countries. We tested cross-national, gender, and age MI using data from nationally representative samples of adolescents using the same research protocol for sampling and data collection methods across fifteen European countries and regions. Our second aim was to test the convergent validity of WHO-5 by exploring its association with other measures of well-being (i.e., psychosomatic complaints, life satisfaction, and self-rated health). 

## 2. Materials and Methods

We used data collected during the 2017/18 Health Behaviour in School-aged Children (HBSC). The HBSC is a World Health Organization collaborative cross-national study that has been conducted every four years since 1983/1984 to monitor the health and well-being of adolescents across Europe and North America. The HBSC uses a standardised research protocol [29]. For each survey round, countries collect data from a nationally representative sample of 11, 13, and 15-year-olds. Stratified random cluster sampling is employed with classes within schools as the primary sampling unit. Adolescents complete anonymous questionnaires in classroom settings. Questionnaires were translated from English into national languages with back-translation checks [29]. Participating countries were eligible to be included in the present analyses if they had collected data using WHO-5, which was an optional measure for the 2017/18 HBSC Survey. This involved analysing data from a total of 79,104 adolescents (50.8% girls; *M*_age_ = 13.54; SD = 1.63) from fifteen countries and regions (a diverse sample with countries from Western Europe, Eastern Europe, and Central Asia). Institutional ethical consent was obtained in each participating country. For a more detailed description of the study sample, see Table 1.

### 2.1. Instruments

*WHO-5 Well-Being Index.* Adolescents were asked to indicate how they felt over the last two weeks for each of the five items. The original items in English were (1) “I have felt cheerful and in good spirits”, (2) “I have felt calm and relaxed”, (3) “I have felt active and vigorous”, (4) “I woke up feeling fresh and rested”, and (5) “My daily life has been filled with things that interest me” [15]. Response options ranged from 0 (“at no time:) to 5 (“all of the time”). In order to undertake the convergent validity analysis, we computed a mean score across the items, where higher values indicated higher mental well-being.

*Psychosomatic complaints.* Adolescents reported how often they experience the following eight health complaints over the past six months: headache, stomachache, backache, feeling low, irritability or bad temper, feeling nervous, difficulties in getting to sleep, feeling dizzy, with the following response options: (1) “about every day”, (2) “more than once a week”, (3) “about every week”, (4) “about every month”, and (5) “rarely or never”. This instrument has adequate test-retest reliability and validity [30]. In the present study, the scale had a sufficient Cronbach’s alpha (0.78). Responses were reversely recoded (0 to 4) so that higher scores indicated more psychosomatic complaints. A mean score was created where a higher score was indicative of higher levels of psychosomatic complaints.

*Life satisfaction* was assessed with the Cantril Ladder [31]. Adolescents rated their life satisfaction on a scale ranging from (0) “The worst possible life” to (10) “The best possible life”. The Cantril ladder has been shown to be a reliable and valid instrument to measure overall mental well-being among adolescents [32].

*Self-rated health* was assessed by a single item. Adolescents were asked to indicate how their health is in general, with the following possible response options: (1) “poor”, (2) “fair”, (3) “good” and (4) “excellen”.

*Gender and age.* Adolescents were asked to indicate whether they are a boy or a girl, as well their date of their birth.

### 2.2. Analytical Approach

As a first step, descriptive statistics (means, standard deviations, and percentages) were examined for gender, age, and WHO-5 score by country. Additionally, means and standard deviations were examined for all items of WHO-5. 

#### 2.2.1. Structural Validity and Reliability

Structural validity defines the extent to which the scores on the scale reflect the underlying dimension [33]. As WHO-5 was developed as a unidimensional scale, we evaluated the factor structure of the scale based on confirmatory factor analysis (CFA) of a one-factor model. We used the following indices and thresholds, in line with the recommendations of Hu and Bentler [34]: comparative fit index (CFI), Tucker-Lewis index (TLI), root mean square error of approximation (RMSEA), and standardised root mean square residual (SRMR) (CFI/TLI = ≥0.90 acceptable, ≥0.95 good; RMSEA = ≤0.08 acceptable, ≤0.06 good; and SRMR = ≤0.10 acceptable, ≤0.08 good). *Reliability* was assessed based on the internal consistency of the scores on the five items using Cronbach’s alpha coefficient. An acceptable value of alpha was set from 0.70 to 0.90 [35]. If alpha is too high, it may suggest that some items are redundant, as they are testing the same underlying construct in a different guise. A maximum alpha value of 0.90 has been recommended [35].

#### 2.2.2. Measurement Invariance 

Measurement invariance of the WHO-5 was assessed using a series (country, gender, and age) of multi-group CFAs. This approach allows for examination of measurement invariance by increasingly constraining parameters to be equal between groups and comparing model results to examine for change [36]. Where little or no change occurs, we can say that the expectation of invariance is met. We ran three models to examine levels of invariance: configural, metric, and scalar. To evaluate the configural model for each group, we used Hu and Bentler’s [34] guidelines for model fit indices: Comparative fit index (CFI) > 0.95; Tucker-Lewis index (TLI) > 0.95; and root mean square error of approximation (RMSEA) < 0.06. If configural invariance was met, we proceeded to examine metric invariance and then scalar invariance models. Comparisons were not made using a Chi-square difference test, as this approach is known to be sensitive to sample size. Instead, given the large sample size in this study, we examined differences between the TLI and CFI values, where a difference ≤ 0.01, and a difference in root mean square error of approximation (RMSEA) ≤ 0.015 indicate invariance [36,37,38,39]. We used robust weighted least squares (WLSMV) estimation as it is suitable for categorical and ordinal data.

Scalar invariance must hold to be able to interpret latent means and correlations across groups. Therefore, if scalar invariance was not fully satisfied, partial measurement invariance was examined by adjusting factor loadings and/or intercept, as it has been argued that meeting partial invariance is sufficient for analysis between groups [36]. Partial invariance is met when the parameters of at least two indicators are equal across groups. More specifically, it aims to identify those items that are very different across groups and release them while making sure that at least two items per latent construct have equal loadings and intercepts. All measurement invariance analysis was conducted in R v. 4.1 using the *lavaan* package.

#### 2.2.3. Convergent Validity

Convergent validity refers to how closely the new scale is related to other variables and other measures of the same construct of the items. In this study, convergent validity was tested by running Pearson correlations between WHO-5 and the health and well-being outcomes listed above (i.e., psychosomatic complaints, life satisfaction, and self-rated health). The correlations were calculated for (i) the aggregated data and (ii) separately for boys and girls. All convergent validity analyses were conducted in Stata 17. 

## 3. Results

### 3.1. Sample Characteristics

In total, 79,104 young people participated in the study with data from 15 European countries. Only those with complete data on the WHO-5 items were included in the analysis (*N* = 74,071; missing *N* = 5033, 6.4%). Table 1 outlines participants’ socio-demographics overall and by country. The WHO-5 items (mean and standard deviation) are outlined in Table 2. The first item (labelled “who1”) showed the highest mean (*M* = 3.34, SD = 1.37) and the item “I woke up feeling fresh and rested” had the lowest mean (*M* = 2.60, SD = 1.68). CFAs within each country were run and are outlined in Appendix A.

### 3.2. Structural Validity and Reliability

CFA models showed that in pooled data across countries, the one-factor model of WHO-5 had a poor model fit (CFI = 0.983; TLI = 0.965; RMSEA = 0.084; and SRMR = 0.020). On average (i.e., in the pooled sample), all factor loadings exceeded 0.50 (Appendix A). More specifically, the WHO-5 showed a good fit only in Armenia, Kazakhstan, and the Republic of Moldova. In the other twelve countries, the one factor model showed a poor model fit (details of the CFA estimated by country can be found in the Appendix A). Internal consistency indices (Cronbach’s alpha) ranged from 0.75 (in Kazakhstan) to 0.92 (in Czechia). In three countries, the alpha coefficient was 0.90 or higher, which denotes that there might be some redundancy in the items in these countries [35].

In the next step, we ran CFA analyses on pooled data and by country using a four-item version of the scale (WHO-4, including the following items: *who2*, *who3*, *who4*, and *who5*). This decision was based on the observation that in the unconstrained model, the error terms of who1 and who2 indicated covariance (outlined below). CFAs on the WHO-4 showed that in pooled data across countries, the one-factor model of the WHO-4 had good model fit (CFI = 0.998, TLI = 0.993, RMSEA = 0.041, and SRMR = 0.009). In almost all countries, the model of WHO-4 showed a good fit. The minimum values were observed in Ukraine (CFI = 0.995, TLI = 0.985, RMSEA = 0.069, and SRMR = 0.012), while the maximum values were observed in the Russian Federation (CFI = 1.000, TLI = 0.999, RMSEA = 0.014, and SRMR = 0.005). The only exceptions were observed for Georgia (CFI = 0.967, TLI = 0.902, RMSEA = 0.190, and SRMR = 0.033), and Czechia (CFI = 0.994, TLI = 0.983, RMSEA = 0.083, and SRMR = 0.012), where the WHO-4 model showed a poor model fit. The overall internal consistency coefficient was 0.86, with country-level values ranging from 0.70 (Kazakhstan) to 0.90 (Czechia). In all countries, factor loadings exceeded 0.50.

### 3.3. Measurement Invariance by Country

Table 3 shows the results of multi-group CFA tests for measurement invariance across countries. The configural model (i.e., the common measurement model), with no equality constraints across countries, did not show an acceptable fit to the data, indicating that the items loaded on different factors for different groups. The modification indices demonstrated covariance between error terms for items 1 and 2. We therefore dropped item 1 (i.e., *who1*). The four-item measure (WHO-4) showed an acceptable model fit. Constraining factor loadings to be equal across countries did not substantially change model fit (Δ_CFI & TLI_ ≤ 0.01; Δ_RMSEA_ ≤ 0.015), showing that latent factors had equivalent relationships with all items across countries and that metric invariance was supported. Having met metric invariance, we tested for scalar invariance by setting intercepts to be equivalent. Scalar invariance was not confirmed as changes in CFI, TLI, and RMSEA were greater than the predefined thresholds. As scalar invariance was not met, we tested for partial invariance by freeing the intercepts of items 2 and 3. In this case, we found that partial invariance was met (see Table 3).

### 3.4. Measurement Invariance by Age and Gender

Table 4 and Table 5 show the results of multi-group CFA tests for measurement invariance across age categories and genders. Similar to the configural model (i.e., the common measurement model), with no equality constraints across countries, these models did not show an acceptable fit to the data. This indicates that the items loaded on different factors for distinct groups for both gender and age, respectively. Consequently, we examined configural model fit after dropping the item *who1*. As seen in the country group model, the four-item measure showed an acceptable configural model fit. Using pre-defined change thresholds, metric and scalar invariance were met for both the gender- and age-grouped models (Table 4 and Table 5, respectively).

### 3.5. Convergent Validity

Overall, WHO-5 (Appendix A) and WHO-4 (Table 6 and Table 7) showed significant medium-effect size correlations in the expected directions, which confirms appropriate convergent validity. The correlation coefficients were slightly stronger for girls than for boys (Table 7).

## 4. Discussion

To our knowledge, the present study is the first to systematically analyse the psychometric properties and measurement invariance of the WHO-5 across comparable nationally representative samples of adolescents across many countries. Our findings from 79,104 young people from fifteen European countries and regions showed that WHO-5 does not show good psychometric properties and measurement invariance. However, by excluding the first item of the scale (“I have felt cheerful and in good spirits”), the WHO-4, consisting of the other four original items, demonstrated good psychometric properties and suitability for cross-national comparisons (as well as age and gender) in adolescent mental well-being.

A study similar to our scope but using samples of adults participating in the European Working Condition Survey 2015 was conducted by Sischka et al. [28]. Overall, they found that the WHO-5 had favourable psychometric characteristics. Their results demonstrated a very good fit in terms of configural and metric invariance while not meeting the criteria for scalar invariance. Therefore, even in this case, there is a limited scope for cross-national and cross-cultural comparisons. Our findings on the lack of scalar invariance echo their findings. However, in our case, the four-item version of the scale met partial measurement invariance.

CFA confirmed good structural validity of the scale across almost all countries, but this was the case only for WHO-4. With a few exceptions, the original scale failed to meet the criteria for acceptable structural validity. This could indicate that the original instrument might show redundancy in the items when used with adolescents. Future studies could use alternative strategies for the evaluation of the structural validity of the scale, such as Rasch analysis [40]. Furthermore, the four-item version showed better internal consistency in all countries than the original WHO-5. Therefore, we can argue that the simplified scale provides reliable scores overall and within each country included in our study.

The factor structure of the original WHO-5 scale did not show an appropriate model fit to indicate measurement invariance. However, by excluding the first item, WHO-4 shows appropriate model fit and supports partial scalar invariance. Other researchers, such as Topp et al. [6] based on a systematic review of 213 articles and Sischka et al. [28] based on a large-sample study with adult participants from the European Working Condition Survey 2015, concluded that the WHO-5 is a psychometrically sound brief measure of subjective well-being. None of their findings suggested that the five-item version of the measure would have been overdetermined, warranting the reduction of the number of items. Our results, on the other hand, demonstrate that the instrument shows better psychometric characteristics—in terms of both internal consistency and MI—if the first item (“I have felt cheerful and in good spirits”) is removed. The error term of this item co-varied with the error term of the second item (“I have felt calm and relaxed”). This indirectly suggests that the concepts of being cheerful and in good spirits and feeling calm and relaxed are not as different in the thinking of adolescents as in adults. While we can only speculate on the reason of this age difference, we need to highlight that subjective well-being differs across age groups: it is lower in middle-aged adults and higher in young people and the elderly [41]. It has been observed in other studies that young people’s non-differentiation of concepts—which adult researchers and subjects would deem different—can increase Cronbach’s alpha, especially if items are worded in the same direction [42,43].

The present results also showcase the need for reflection on whether measures developed to assess subjective well-being in adults, irrespective of length, are suitable for younger age groups. While the one-item difference between WHO-5 and WHO-4 may not seem essential, given the attentional span of young people and the sometimes-overburdening length of questionnaires administered to them, even omitting one item might reduce the time participants spend filling them out. Researchers also need to bear in mind that the composition and wording of the questionnaire on its own conveys an indirect message to adolescents on how one should live. Some young people experience this message as positive, but in others it evokes feelings of shame [44]. 

The main strength of the present study is that it uses data from the HBSC, a well-established survey of nationally representative samples of adolescents aged 11 to 15, from various countries and regions. However, there are a series of limitations which warrant acknowledgement. First, we used a cross-sectional sample, which limits temporal comparisons. Second, since the WHO-5 scale was employed in the national HBSC studies at the discretion of the local research teams, we only had data from a limited number of countries. Third, despite the rigorous translation-backtranslation procedures in HBSC [29] and the fact that the WHO-5 already had established versions in many countries’ native or first language, it remains a question whether all of these versions are functionally and semantically equivalent to each other.

## 5. Conclusions

Cross-country comparisons using the WHO-5 can advance our understanding of adolescent mental health and mental well-being. This is a freely available, short instrument that uses only positively worded items and can be embedded in school-health surveys. Furthermore, this scale can inform and drive global and national intervention and prevention efforts in the area of adolescent mental health and well-being. The present study introduces a revised version of the WHO-5, the WHO-4, that allows for a valid comparison of mental well-being across 15 countries across different regions in Europe. The WHO-4 proved to be a reliable and valid instrument for use with adolescent populations. 

## Figures and Tables

**Table 1 ijerph-19-09798-t001:** Demographics, WHO-5 score, and sample size by country (*N* = 74,071).

Country	*n*	Girls (%)	Age (*M*, SD)	WHO-5 (*M*, SD)	Missing (%)
Austria	3988	50.95	13.26 (1.62)	2.75 (1.19)	0.10
Armenia	4374	50.94	13.53 (1.64)	3.46 (1.16)	2.35
Czechia	9813	49.91	13.47 (1.65)	2.92 (1.30)	12.20
Georgia	3856	51.48	13.45 (1.72)	3.01 (1.41)	6.27
Ireland	3636	49.34	13.42 (1.57)	2.98 (1.18)	2.03
Kazakhstan	4354	50.39	13.30 (1.69)	3.28 (1.07)	1.50
Lithuania	3710	49.89	13.71 (1.65)	3.05 (1.08)	0.21
Republic of Moldova	4476	50.45	13.56 (1.67)	3.31 (1.06)	0.17
Poland	5127	51.00	13.60 (1.66)	2.76 (1.09)	0.25
Romania	4392	51.43	13.22 (1.63)	3.09 (1.15)	0.46
Russian Federation	4104	52.61	13.82 (1.66)	3.03 (1.10)	1.57
Slovenia	5540	49.51	13.59 (1.63)	2.97 (1.14)	0.23
Scotland	4852	52.14	13.52 (1.63)	2.88 (1.14)	0.72
Turkey	5540	51.45	13.44 (1.71)	2.54 (1.21)	0.62
Ukraine	6178	52.23	13.42 (1.63)	3.18 (1.28)	3.90
Overall	74,071	50.88	13.48 (1.66)	3.00 (1.21)	6.36

**Table 2 ijerph-19-09798-t002:** Description of items (*N* = 74,071).

Item Label	Item Description	*M*	SD
who1	I have felt cheerful and in good spirits	3.32	1.37
who2	I have felt calm and relaxed	2.93	1.46
who3	I have felt active and vigorous	3.17	1.49
who4	I woke up feeling fresh and rested	2.60	1.68
who5	My daily life has been filled with things that interest me	2.98	1.56

**Table 3 ijerph-19-09798-t003:** Measurement invariance testing by country (*N* = 74,071).

Models	df	CFI ^a^	TLI ^a^	RMSEA ^a^	ΔCFI	ΔTLI	ΔRMSEA	Model Comparisons
**WHO-5**								
1. Configural	75	0.982	0.964	0.073	—	—	—	—
**WHO-4 ^b^**								
1. Configural	30	0.996	0.989	0.045	—	—	—	—
2. Metric	72	0.989	0.986	0.049	**−0.007**	**−0.003**	**0.004**	2 vs. 1
3. Scalar	114	0.952	0.962	0.082	−0.037	−0.024	0.033	3 vs. 2
4. Partial Invariance ^c^	86	0.980	0.979	0.061	**−0.009**	**−0.007**	**0.012**	4 vs. 2

^a^ Robust estimates are reported. ^b^ Item 1 was excluded. ^c^ Intercepts of items 2 and 3 are freed of constraints. Bold values indicate invariance is met. CFI = comparative fit index; TLI = Tucker Lewis Index; RMSEA = root mean square error of approximation.

**Table 4 ijerph-19-09798-t004:** Measurement invariance testing by gender (*N* = 74,071).

Models	df	CFI ^a^	TLI ^a^	RMSEA ^a^	ΔCFI	ΔTLI	ΔRMSEA	Model Comparisons
**WHO-5**								
1. Configural	10	0.983	0.967	0.069	—	—	—	—
**WHO-4 ^b^**								
1. Configural	4	0.998	0.994	0.033	—	—	—	—
2. Metric	7	0.998	0.997	0.023	**0.000**	**0.003**	**−0.010**	2 vs. 1
3. Scalar	10	0.994	0.993	0.035	**−0.004**	**−0.004**	**0.012**	3 vs. 2

^a^ Robust estimates are reported. ^b^ Item 1 was excluded. Bold values indicate invariance is met. CFI = comparative fit index; TLI = Tucker Lewis Index; RMSEA = root mean square error of approximation.

**Table 5 ijerph-19-09798-t005:** Measurement invariance testing by age category (*N* = 73,816).

Models	df	CFI ^a^	TLI ^a^	RMSEA ^a^	ΔCFI	ΔTLI	ΔRMSEA	Model Comparisons
**WHO-5**								
1. Configural	15	0.981	0.963	0.071	—	—	—	—
**WHO-4 ^b^**								
1. Configural	6	0.997	0.992	0.037	—	—	—	—
2. Metric	12	0.996	0.994	0.030	**−0.001**	**0.002**	**−0.007**	2 vs. 1
3. Scalar	18	0.987	0.987	0.045	**−0.009**	**−0.007**	**0.015**	3 vs. 2

^a^ Robust estimates are reported. ^b^ Item 1 was excluded. Bold values indicate invariance is met. CFI = comparative fit index; TLI = Tucker Lewis Index; RMSEA = root mean square error of approximation.

**Table 6 ijerph-19-09798-t006:** Convergent validity: correlation coefficients using aggregated data (WHO-4) (*N* = 74,071).

	1	2	3	4
1. WHO-4 Well-being	1			
2. Psychosomatic complaints	−0.445	1		
3. Life satisfaction	0.431	−0.376	1	
4. Self-rated health	−0.335	0.334	−0.371	1

All correlations are significant at *p* < 0.05.

**Table 7 ijerph-19-09798-t007:** Convergent validity by gender: correlation coefficients using aggregated data (*N* = 74,071).

	1(Boys)	2 (Boys)	3 (Boys)	4 (Boys)
1. WHO-4 Well-being (girls)	1	**−0.376**	**0.385**	**−0.292**
2. Psychosomatic complaints (girls)	−0.487	1	**−0.319**	**0.276**
3. Life satisfaction (girls)	0.467	−0.417	1	**−0.349**
4. Self-rated health (girls)	−0.362	0.382	−0.386	1

All correlations are significant at *p* < 0.05. To facilitate reading, the values for boys are in bold.

## Data Availability

Data available upon request to the contact author.

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
