# Peer review of "Measurement Invariance of the WHO-5 Well-Being Index: Evidence from 15 European Countries"

_ijerph, 2022, doi:10.3390/ijerph19169798_

Round 1
Reviewer 1 Report
line 275 - typo "Iitem 1"
line 358 - change "irrespective of how their length" to "irrespective of length"
line 484 - Lundegard didn't print properly on the pdf. I'm not sure if this is a problem only on my end.
Well done. WHO-4 is a much needed tool for cross-country comparisons for adolescents.
Author Response
LIST OF CHANGES- ijerph-1857435
Resubmission August 2022
Reviewer #1
R1.1 line 275 - typo "Iitem 1"
RESPONSE:
We would like to thank the reviewer for the positive appraisal of our manuscript. We hope indeed that this will inform future studies planning to use WHO-5, or WHO-4, in cross-national research.
R1.2 line 358 - change "irrespective of how their length" to "irrespective of length"
RESPONSE:
Thank you for this observation. This has been fixed.
R1.3 Lundegard didn't print properly on the pdf. I'm not sure if this is a problem only on my end.
RESPONSE:
Thank you for this observation. It seems to have been a typo due to the metadata used by the reference manager we used. This has been fixed.
R1.4 Well done. WHO-4 is a much needed tool for cross-country comparisons for adolescents.
RESPONSE:
We would like to thank the reviewer for the positive appraisal of our manuscript. We hope indeed that this will inform future studies planning to use WHO-5, or WHO-4, in cross-national research.
Reviewer 2 Report
The study tackles an interesting problem, and it provides an actionable solution based on the findings. Hopefully, others will take it from here and suggest a good replacement for the first statement of WHO-5.
The article is easy to read, and it appears to be technically correct.
It would help to clarify how questionnaires were collected. Currently, it says "Adolescents complete anonymous questionnaires in classroom settings." If they were paper-based, it might be enough to add the description after the 3rd word.
It might be interesting for a reader to learn also whether respondents complained about the 0-5 Likert scale, i.e. because of not having a middle value. (Esp. in case if they were paper-based questionnaires, so there was somebody in the classroom during data collection.)
There are several minor issues:
It is advisable to discuss with the journal editor how to address the citation of
"The World Health Organization (WHO) defines mental health as “a state of well-being in 40
which the individual realizes his or her own abilities, can cope with the normal stresses 41
of life, can work productively and fruitfully, and is able to make a contribution to his or 42
her own community” (2016, para 2)."
The reference is possibly missing in the list of references.
Another citation issue that needs to be fixed is
This instrument was origi- 59
nally developed as a measure of mental well-being for adult populations (Balázs et al., 60
2018; McMahon et al., 2017)[6].
Two instances of "(Inchley et al., 2018)" probably need to be changed to [28].
There is probably an extra I in "The first item (ilabelled who1)".
It would be worth to think about labeling of the first column in Tables 3-5. E.g. could WHO-5/WHO-4 be moved lower - merging it with the name of models? Or could there be a thick (instead of a thin) line between Configural and WHO-4b?
In table S1, there is obviously an extra 0 in max loading for Poland (it should be 0.75 instead of 0.075).
Author Response
LIST OF CHANGES- ijerph-1857435
Resubmission August 2022
Reviewer #2
R2.1 The study tackles an interesting problem, and it provides an actionable solution based on the findings. Hopefully, others will take it from here and suggest a good replacement for the first statement of WHO-5.
The article is easy to read, and it appears to be technically correct.
RESPONSE:
We would like to thank the reviewer for the positive appraisal of our manuscript. We hope indeed that this will inform future studies planning to use WHO-5, or WHO-4, in cross-national research.
R2.2 It might be interesting for a reader to learn also whether respondents complained about the 0-5 Likert scale, i.e. because of not having a middle value. (Esp. in case if they were paper-based questionnaires, so there was somebody in the classroom during data collection.)
RESPONSE:
We thank the reviewer for raising this issue. Indeed, the Likert scale used does not have a middle value. We want to inform the reviewer that the HBSC survey uses a variety of questions; some use a classic Likert scale (either Agree – Disagree; 1 to 5 or 1 to 7; all have a middle value), some use a likert scale that does not have a middle value (such as WHO-5 Wellbeing index or Psychosomatic complaints), and questions that asks about the frequency with which they engage in certain behaviours (these are measured in days or times). The HBSC research protocol requires that items are validated before included in the survey (including qualitative validation) and countries need to undergo a national pilot study of the survey for each HBSC round in which they test the length of the questionnaire and ask for qualitative feedback regarding the questions asked (including the response scale used). With regards to this specific issue raised (ie., questions that do not have a middle point) there have been no issues reported during these pre-survey phases outlined above.
R2.3 There are several minor issues:
It is advisable to discuss with the journal editor how to address the citation of
"The World Health Organization (WHO) defines mental health as “a state of well-being in 40
which the individual realizes his or her own abilities, can cope with the normal stresses 41
of life, can work productively and fruitfully, and is able to make a contribution to his or 42
her own community” (2016, para 2)."
The reference is possibly missing in the list of references.
RESPONSE:
Thank you for this observation. We have updated the reference which is not linked in the revised manuscript (see reference no #4).
R2.4 Another citation issue that needs to be fixed is
This instrument was origi- 59
nally developed as a measure of mental well-being for adult populations (Balázs et al., 60
2018; McMahon et al., 2017)[6].
RESPONSE:
Thank you for this observation. This has been fixed.
R2.5 Two instances of "(Inchley et al., 2018)" probably need to be changed to [28].
RESPONSE:
Thank you for this observation. This has been now corrected.
R2.6 There is probably an extra I in "The first item (ilabelled who1)".
RESPONSE:
Thank you for this observation. This has been now corrected.
R2.7 It would be worth to think about labelling of the first column in Tables 3-5. E.g. could WHO-5/WHO-4 be moved lower - merging it with the name of models? Or could there be a thick (instead of a thin) line between Configural and WHO-4b?
RESPONSE:
Thank you for this observation. We have indeed struggled a bit with formatting the tables. We made some amendments by inserting a new row. Hopefully this reads better.
R2.8 In table S1, there is obviously an extra 0 in max loading for Poland (it should be 0.75 instead of 0.075).
RESPONSE:
Thank you for this observation. This has been now corrected.